# Prevalence and Predictors of Pressure Injuries in Patients with Spincal Cord Injuries Using Clinical Data

Lindsay Stern
Institue of Biomedical Engineering; KITE Research Institute
University of Toronto; University Health Network
*Toronto, Canada*
lindsay.stern@mail.utoronto.ca

Atena Roshan Fekr
Institue of Biomedical Engineering; KITE Research Institute
University of Toronto; University Health Network
*Toronto, Canada*
atena.roshanfekr@utoronto.ca

*Abstract*— **Pressure injuries (PIs) are injuries to the skin and/or underlying tissue, typically along the bony prominences of the body, often caused by pressure and shear forces. People with spinal cord injuries (SCIs) have a 25%-85% lifetime risk of developing pressure injuries due to various comorbidities, including neurological and musculoskeletal challenges. Currently, risk assessment scales, such as the Braden Scale or Spinal Cord Injury Pressure Ulcer Scale are used to define the risk of developing PIs. The scales are often used to aid clinicians in developing a care plan for PI prevention. However, there is a lack of consensus on which scale is best to use, often requiring clinical best judgment and thus creating a subjective assessment. Studies have indicated that clinical data, such as levels of hemoglobin, lymphocytes, or creatine level, may offer a more objective assessment for defining PI risk. Albeit useful, these studies often offer a generalized conclusion which does not consider the unique case of patients with SCIs. Therefore, this study explores demographic and clinical profile of individuals with SCIs to identify predictors associated with the development of PIs. The demographic data highlights a decrease in mean age and BMI amongst patients with PIs, but an increase in patients' length of stay in hospital compared to those without PIs. The clinical data highlights differences in multiple biomarkers, including creatinine, cholesterol, and hemoglobin levels. Therefore, these results highlight the unique demographic and clinical predictors that can be leveraged to build Artificial Intelligent models for early, objective prediction of PIs in patients with SCIs. In turn, this will aid clinicians in developing an appropriate care plan for patients and thus further reduce PIs development.**

*Keywords*— *Pressure Injury, Spinal Cord Injury, Prediction, Risk Factors*

## I. INTRODUCTION

Pressure injuries (PIs) can be defined as injuries to the skin and/or underlying tissue caused by prolonged pressure and shear forces, often occurring along the bony prominent regions of the body [1]. Some risk factors for developing PIs include aging, decrease in mobility, decrease in nutritional health, decrease in sensory perception, and changes to skin characteristics [2]. In Canada, 26% of Canadians across all healthcare settings develop hospital acquired pressure injuries (HAPIs), with treatment ranging between $1,247 to $597,363 CAD per patient [2]. In the United States, 3 million adults acquire HAPIs, with treatment ranging from $500 to $70,000 USD per patient [3]. More specifically, people with spinal cord injuries (SCIs) are at higher risk of developing PIs, with studies indicating a lifetime prevalence ranging from 25%-85% [4][5]. People with SCIs are more susceptible to developing PIs due to common comorbidities, including cardiovascular, neurological, and musculoskeletal challenges [5].

Currently, there are multiple risk assessment scales to aid clinicians in defining patient risk levels of developing PIs. These scales include the Braden Scale [6], Norton Pressure Sore Risk-Assessment Scale [7], Waterlow Pressure Ulcer Scale [8], and Spinal Cord Injury Pressure Ulcer Scale (SCIPUS) [9]. Each scale evaluates various factors, with some overlaps. A study comparing SCIPUS to the Braden Scale discovered that SCIPUS achieves higher sensitivity in identifying SCI patients at risk of PIs [10]. It is evident that there are a multitude of factors that may influence patient risk of PIs development, with no clear consensus as to which scale is better. According to best clinical practice, it is recommended that these scales are used in conjunction with clinicians' best judgement which results in a subjective assessment [11]. In reviewing studies for a more objective assessment, Clinical Practice Guideline found that among 14 studies reviewed [11], 56% of studies highlighted low hemoglobin (reducing oxygen supply to tissues) as a significant factor in PIs development. Both studies that examined lymphocyte (>1.5) also found a positive correlation with PI development. Studies also showed that creatine level is associated with risk of developing PIs [12][13][14]. One study determined that the best health data predictors of all stages of PIs include the patient's BMI, age, surgical time, hemoglobin, and creatine levels [15]. Reese et al. illustrated that in a cohort of 197,991 patients, 5458 developed HAPIs from January 2018 to July 2022 [16]. This study determined that top five important characteristics of predicting PIs include tracheostomy, edema, central line, first albumin measure, and age [16]. However, these studies often generalize the findings, not taking into consideration the unique case of people with SCIs. Therefore, this paper will examine and compare demographic and health characteristics of people with SCIs who have and have not developed HAPIs.

The subsequent sections are organized as follows: Section II the dataset and methodology used, Section III describes the experimental results, Section IV illustrates experimental observations and limitations, and Section V summarizes concluding remarks.

## II. DEMOGRAPHIC DATASET DESCRIPTION & ANALYSIS

The dataset analyzed within this study was approved by the Research Ethics Board under the University Health Network in Toronto. Electronic Health Record (EHR) were extracted from the hospital database covering the period from January 2019 to September 2024. In total, 158 patients with SCIs were identified with 24.05% patients (n=38) diagnosed with pressure injuries.

Mitacs Accelerate Funding: IT40644

TABLE I. DATASET DEMONGRAPHIC CHARACTERISTICS

|  | With PIs (n=38) | Without PIs (n=120) |
|---|---|---|
| **Sex, n (%)** |  |  |
| Male | 28 (73.68) | 79 (65.83) |
| Female | 10 (26.32) | 41 (34.17) |
| **Mean Age** | 62.4 ± 15.72 | 64.67 ± 17.68 |
| **Mean BMI** | 26.73 ± 7.42 | 28.33 ± 5.79 |
| **Length of Stay (days)** | 1-62 | 1-21 |

TABLE II. BRADEN SCALE RESULTS FOR 158 PATIENTS

|  | With PIs (n=38) | Without PIs (n=120) |
|---|---|---|
| Very High Risk, n (%) | 1 (2.63) | 0 (0.00) |
| High Risk, n (%) | 9 (23.68) | 4 (3.33) |
| Moderate Risk, n (%) | 11 (28.95) | 14 (11.67) |
| Mild Risk, n (%) | 7 (18.42) | 47 (39.17) |
| No Risk, n (%) | 2 (5.26) | 22 (18.33) |
| Not Assessed | 8 (21.05) | 34 (27.50) |

TABEL III. RESSURE INJURY STAGE BREAKDOWN

| PI Stage | Cohort Quantity (%) |
|---|---|
| Stage I PI | 5.26 |
| Stage II PI | 21.05 |
| Stage III PI | 23.68 |
| Stage IV PI | 28.95 |
| Unstageable PI | 23.68 |
| Suspected DTI | 7.89 |
| PI, No Stage Identified | 23.68 |

In this hospital, patients are typically assessed for PIs risk using the Braden Scale. Table II presents the risk assessment results for patients with and without PIs. These results highlight that the majority of patients (55.26%) who developed PIs were classified as being at *moderate* to *very high* risk, whereas most patients who did not develop PIs were categorized as having *mild* risk. It is important to note that, 41 patients had no recorded Braden Scale assessment which may suggest that appropriate care plans may not have been established. There are six stages of PIs, ranging from a Stage I PI with no open wound to deep tissue injury (DTI) [17]. Table III highlights the PI staging breakdown of the cohort. These results highlight that Stage IV PI was the most prevalent among the cohort, accounting for 28.95% of cases. This stage indicates full-thickness tissue loss with exposure of muscle, tendon, or bone [17] highlighting the urgent need for improved risk assessment tools to further prevent PI development.

## III. CLINICAL DATASET DESCRIPTION & ANALYSIS

Clinical data, such as patient total surgery time, blood type, and vital signs were extracted for patients with and without PI development, with results shown in Table IV. These results highlight that on average, patients with PIs have a lower duration of surgery, more commonly have blood type A (compared to those without PIs who more commonly have blood type O), and have slightly higher pulse and respiratory rates.

TABLE IV. CLINICAL DATA DESCRIPTION, INCLUDING SURGERY DURATION, BLOOD TYPE, AND VITAL SIGNS

|  | With PIs (n=38) | Without PIs (n=120) |
|---|---|---|
| **Blood Type, n (%)** |  |  |
| O | 10 (26.32) | 44 (36.67) |
| A | 16 (42.11) | 28 (23.33) |
| B | 3 (7.89) | 22 (18.33) |
| AB | 0 (0.00) | 9 (7.50) |
| Not Documented | 9 (23.68) | 17 (14.17) |
| **Mean Vital Signs** |  |  |
| O2 Stauration (%) | 96.50 ± 1.61 | 96.56 ± 1.12 |
| Pulse (bpm) | 80.90 ± 12.55 | 78.89 ± 10.12 |
| Resperatory Rate (breaths/minute) | 18.0 ± 1.27 | 17.64 ± 1.20 |
| Temperature (°C) | 36.61 ± 0.24 | 36.66 ± 0.21 |

The mean and standard deviation of blood lab values for patients with and without PIs are presented in Table V and normalized values are shown in Fig. 1.

These results highlight that there are often differences in blood lab values between these two groups. As the data was not normally distributed, a Mann-Whitney U statistical analysis was used to compare the blood lab values between patients with PIs and those without. Notably, the cholesterol and glomerular filtration rate (GFR) blood levels statistically

TABLE V. MEAN BLOOD LAB VALUES AND ASSOCIATED P-VALUES

|  | With PIs (n=38) | Without PIs (n=120) | p-value |
|---|---|---|---|
| Albumin (g/L) | 30.43 ± 5.08 | 32.43 ± 5.89 | 0.10 |
| Alkaline Phosphate (U/L) | 115.13± 48.10 | 107.79 ± 68.65 | 0.06 |
| Alanine Transaminase (U/L) | 45.42 ± 43.03 | 41.07 ± 48.22 | 0.47 |
| Amylase (U/L) | 52.17 ± 29.28 | 63.00 ± 42.66 | 0.18 |
| Aspartate Aminotransferase (U/L) | 41.33 ± 34.76 | 44.32 ± 55.04 | 0.46 |
| Basophils (x10e9/L) | 0.04 ± 0.03 | 0.03 ± 0.03 | 0.40 |
| Beta 2 Microglobulin (mg/L) | 4.80 ± 0.00 | 1.60 ± 0.00 | 0.19 |
| **Bilirubin (umol/L)** | **10.91 ± 5.47** | **13.77 ± 8.89** | **0.0467** |
| Calcium (mmol/L) | 1.85 ± 0.27 | 1.77 ± 0.40 | 0.17 |
| Chloride (mmol/L) | 103.99 ± 3.89 | 104.16 ± 3.11 | 0.36 |
| **Cholesterol (mmol/L)** | **2.49 ± 0.91** | **2.26 ± 0.95** | **0.0412** |
| Creatine Kinase (U/L) | 190.44 ±15.27 | 532.38 ±23.67 | 0.16 |
| **Creatinine (umol/L)** | **69.70 ± 36.61** | **81.59 ± 96.47** | **0.0085** |
| Eosinphil (x10e9/L) | 0.19 ± 0.16 | 0.17 ± 0.15 | 0.17 |
| **GFR (mL/min/1.72m2)** | **98.99 ± 22.13** | **92.84 ± 21.46** | **0.0038** |
| Glucose (mmol/L) | 6.60 ± 1.36 | 7.47 ± 2.56 | 0.07 |
| **Hemoglobin (g/L)** | **103.03 ±16.09** | **117.20 ±17.07** | **<0.0001** |
| Lactate (mmol/L) | 1.76 ± 0.97 | 1.81 ± 0.97 | 0.43 |
| Lipase (U/L) | 27.78 ± 33.15 | 27.78 ± 40.02 | 0.11 |
| Lymphocytes (x10e9/L) | 1.68 ± 1.26 | 1.43 ± 0.79 | 0.17 |
| Magnesium (mmol/L) | 0.82 ± 0.07 | 0.83 ± 0.09 | 0.36 |
| Monocytes (x10e9/L) | 0.70 ± 0.24 | 0.79 ± 0.40 | 0.06 |
| Neutrophils (x10e9/L) | 7.47 ± 3.22 | 7.27 ± 2.18 | 0.31 |
| Phosphate (mmol/L) | 1.02 ± 0.17 | 1.04 ± 0.16 | 0.47 |
| Potassium (mmol/L) | 3.95 ± 0.30 | 3.96 ± 0.31 | 0.24 |
| Protein (g/L) | 64.93 ± 10.15 | 61.44 ± 9.32 | 0.06 |
| Sodium (mmol/L) | 138.35 ± 3.06 | 138.03 ± 4.23 | 0.37 |
| Triglycerides (mmol/L) | 1.94 ± 0.92 | 1.42 ± 0.69 | 0.08 |
| Troponin I (ng/L) | 139.28 ±75.72 | 283.16 ±95.16 | 0.40 |

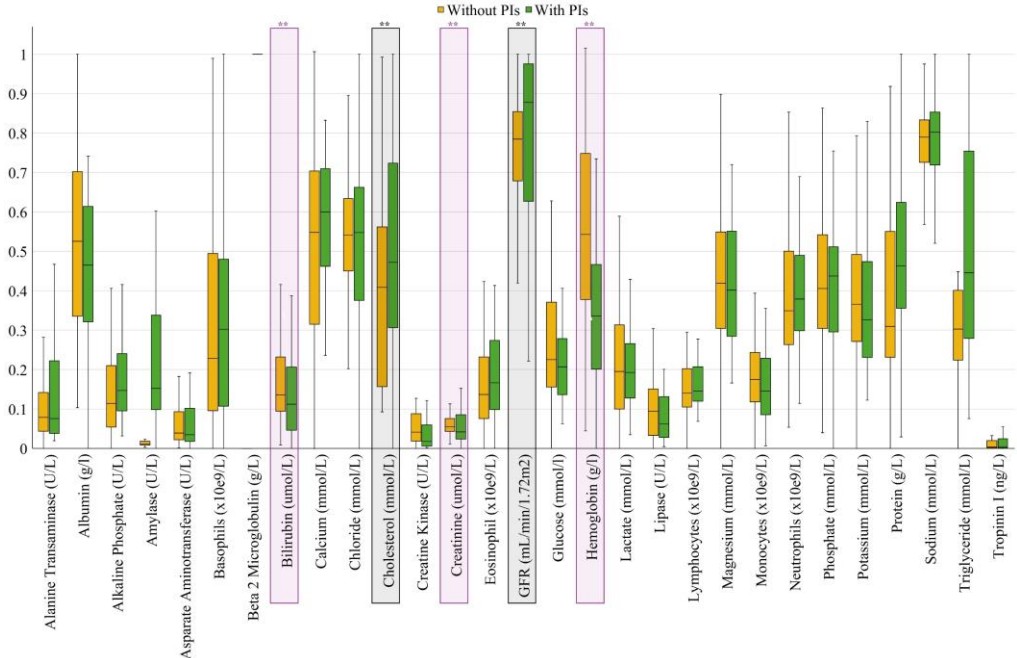

Fig. 1. The normalized mean values of the blood lab values for patients with PIs (green) and patients without PIs (yellow). The black ** highlight the values that are statistically higher in the PI cohort and the purple ** highlights the values that are staistically lower in the PI cohort. Note: outliers have been removed from this image.

higher in patients with PIs when compared to patients without PIs. Contrastingly, bilirubin, creatinine, and hemoglobin blood levels were statistically lower in patients with PIs compared to those without PIs. These findings align with the previous literature, found in [12][13][14][15].

Tables VI and VII highlight the ten most common associated comorbidities for patients with and without PIs. From these results, it can be shown that both cohorts commonly are diagnosed with benign hypertension, indicating high blood pressure. However, the PI cohort includes both physical diagnoses, such as urinary tract

TABLE VI. TEN COMMON COMORBIDITIES - PATIENTS WITH PIs.

|  | PI Cohort Quantity, n (%) |
|---|---|
| Urinary Tract Infection | 23 (60.53) |
| Benign Hypertension | 18 (47.37) |
| Sepsis | 16 (42.11) |
| Escheria Coli [E. coli] | 15 (39.47) |
| Anaemia | 12 (31.58) |
| Neuromuscular Dysfunction of Bladder | 12 (31.58) |
| Constipation | 11 (28.95) |
| Depressive Episode | 11 (28.95) |
| Hypotension | 11 (28.95) |
| Pneumonia | 11 (28.95) |

TABLE VII. TEN COMMON COMORBIDITIES - PATIENTS WITHOUT PIs.

|  | Non-PI Cohort Quantity, n (%) |
|---|---|
| Benign Hypertension | 61 (50.83) |
| Abnormal Reaction/Complications due to Artifical Internal Device | 35 (29.17) |
| Unspecified Fall | 33 (27.50) |
| Myelopathy | 29 (24.17) |
| Arthrodesis | 23 (19.17) |
| Gastro-oesophageal Reflux | 20 (16.67) |
| Type II Diabetes | 19 (15.83) |
| Disorder of Lipoprotein Metabolism | 17 (14.17) |
| Haemorrhage & Haematoma Complicating Procedure | 17 (14.17) |
| Atrial Fibrillation | 16 (13.33) |

infection, sepsis, or anemia, as well as mental health diagnoses, such as depressive episodes.

## IV. DISCUSSION

This analysis reveals clear demographic and clinical differences between patients with SCIs who develop PIs and those who do not. Noticeable differences are particularly observed in blood lab values, including creatinine, cholesterol, and hemoglobin levels. While some of these findings are aligned with existing literature [12][13][14][15], others may be specific to the SCI population.

Although these results are promising, it is important to recognize the limitations of our work. Firstly, race or ethnicity is not well documented within the hospital network and was therefore not provided to the researchers by the data extraction team. Literature has indicated that people with darker skin pigmentation are often at higher risk of developing PIs [18] [19]. Therefore, this demographic characteristic is likely to be an important factor when predicting the development of PIs. Secondly, the data analysed within this paper were mean results over all hospitalization instances for all patients, summarizing patients' overall clinical profiles. However, these results did not explore time-related trends, a future consideration, which may reveal early signs of pressure injury development or highlight the effects of some preventative measures. Lastly, there results did not analyze the clinical data prior to and post PI development. Clinical results prior to PI diagnosis are important to consider when predicting the potential of PI development. Therefore, in the future, we will analyze time series before and after PIs diagnosis to develop a more robust predictive model for PI risk in patients with SCIs.

## V. CONCLUSION

PIs prevalence as well as demographic and clinical factors were examined from a cohort of 158 patients with spinal cord injuries. These results highlight 24.05% of the population developed PIs, with 28.95% developing a Stage IV PI,

indicating injury to the muscle, tissue, or bone. These results further highlight the need for a more objective PIs risk assessment tool to further guide clinicians in developing effective care plans for all patients. Demographic and clinical data indicate that there are differences between patients who develop and do not develop PIs. These differences include BMI, length of stay in hospital, and various blood lab values (i.e. creatinine, cholesterol, and hemoglobin). These results highlight the potential of developing a machine learning or deep learning algorithm to predict PIs development, ultimately offering a more objective assessment of PIs risk for clinicians to use during care.

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
