# OpenReview forum: "Prevalence and Predictors of Pressure Injuries in Patients with Spinal Cord Injuries using Clinical Data"
_IEEE.org/EMBS/BHI/2025/Conference — BHI 2025_

### Official Review · Reviewer_bjsc · 2025-07-06
**This paper investigates clinical and demographic predators for identifying pressure injuries (PI) development in a small cohort of spinal cord injury patients, establishing the groundwork for machine learning tools that facilitate early detection of PI and improve treatment outcomes.**

**Confidence:** 5
**Clarity Of Writing:** great
**Clinical Significance:** good
**Methodological Novelty:** good
**Overall Rating:** 7

**Experiments And Results:**

great

**Questions For The Authors:**

In the abstract, the use of word "factor" is confusing, please correct it if necessary.
The concept of statistically lower and higher is not clear based on the Figure 1.
Please follow the appropriate format for references to maintain uniformity.

**Strengths:**

This work focused on thorough analysis for finding statistically significant predators based on demographic and clinical data in SCI patients. And verified clinical predators for SCI patients based on Table V. and Fig. 1 aligns with predators reported for generalized patients. Breakdown different stages of PI for indicating the importance of finding the predators, which could facilitate in early detection of PI to prevent reaching higher stage of PI in SCI patients.

**Summary Of The Paper:**

This paper addresses the direction of building machine learning models for significant burden of pressure injuries (PI), which affect 24% of hospitalized patients in Canada and 3 million adults in the US, resulting in substantial treatment costs. While existing quantitative PI assessment tools provide generalized conclusions from all categories of patients, this study focuses specifically on spinal cord injury (SCI) patients, who have a lifetime PI prevalence of 25% to 85%. Unlike previous work that examined clinical and demographic predictors in large general cohorts with less than 2.5% PI prevalence, this research concentrates on 158 SCI patients, of whom 24.05% (n=38) were diagnosed with PI. The paper thoroughly analyzes the demographic and clinical characteristics of these 158 SCI patients and demonstrates how specific clinical and demographic predators can potentially identify PI onset in this population.

**Weaknesses:**

This study focused on a small cohort of SCI patients from a single institution, which may limit the statistical significance of the predictors.

---

### Official Review · Reviewer_yBaz · 2025-07-15
**Analyzing whether sex effect was associated with pressure injury**

**Confidence:** 3
**Clarity Of Writing:** good
**Clinical Significance:** fair
**Methodological Novelty:** fair
**Overall Rating:** 4
**Final Rating:** 5

**Experiments And Results:**

good

**Questions For The Authors:**

- You report the sex distribution in your cohort, but did you analyze whether sex was associated with pressure injury development? Given the observed imbalance, how did you account for potential gender-related effects or bias in your results?
- How might the use of average lab values over multiple hospitalizations obscure important time-related trends?

**Strengths:**

Including clear, well-organized tables comparing patients with and without PIs across demographics, lab values, and comorbidities.

**Summary Of The Paper:**

This paper analyzes hospital records from 158 patients with spinal cord injuries, comparing those who developed pressure injuries to those who did not. It examines differences in demographics, vital signs, blood lab values, and comorbidities, identifying factors such as lower hemoglobin and higher cholesterol in patients with PIs. The study suggests these findings could inform the development of improved clinical risk assessment tools.

**Weaknesses:**

- The Discussion section is too brief, with a significant portion focused only on limitations.
- The paper does not address how these findings might generalize to other settings or populations, limiting the broader applicability and validation of the results.
- The analysis is limited to descriptive statistics and bivariate comparisons, without any multivariate analysis or predictive modeling to identify independent predictors.

---

### Official Review · Reviewer_FjY3 · 2025-07-15
**Prevalence and Predictors of Pressure Injuries in Patients with Spinal Cord Injuries using Clinical Data - Review**

**Confidence:** 5
**Clarity Of Writing:** excellent
**Clinical Significance:** excellent
**Methodological Novelty:** excellent
**Overall Rating:** 8

**Experiments And Results:**

excellent

**Questions For The Authors:**

Satisfactory.

**Strengths:**

[1] The paper proposed using Artificial Intelligence models to determine early, objective prediction of PIs risk in patients with SCIs, highlighting a forward-looking, practical application for personalised patient care.
[2] The paper fills a gap in existing literature that often overlooks the high-risk subgroup by focusing specifically on spinal cord injury patients.
[3] The paper demonstrates statistically significant biomarkers such as hemoglobin, creatinine, and cholesterol as predictors of pressure injuries, which can be used to support the development of objective assessment tools.

**Summary Of The Paper:**

The paper investigates the prevalence and predictors of pressure injuries (PIs) in patients with spinal cord injuries (SCIs), finding that 24.05% of the cohort developed PIs, with 28.95% developing a Stage IV PI. The paper shows that differences in demographic and clinical factors such as BMI, length of hospital stay, and blood biomarkers like haemoglobin, cholesterol, and creatinine levels can be leveraged to build Artificial Intelligence models for early, objective prediction of PIs in patients at risk.

**Weaknesses:**

[1] Some patients lacked Braden Scale assessments, which may introduce bias or incomplete risk profiling into the dataset.
[2] There is no validation of the use of predictive models in this paper, making the proposed AI application a theoretical suggestion rather than a tested outcome.